# Antimicrobials Functioning through ROS-Mediated Mechanisms: Current Insights

**DOI:** 10.3390/microorganisms10010061

**Published:** 2021-12-28

**Authors:** Ankita Vaishampayan, Elisabeth Grohmann

**Affiliations:** Department of Microbiology, Faculty of Life Sciences and Technology, Berlin University of Applied Sciences, Seestrasse 64, 13347 Berlin, Germany; egrohmann@bht-berlin.de

**Keywords:** oxidative stress, reactive oxygen species, antimicrobials, antibiotic resistance

## Abstract

Antibiotic resistance and infections caused by multidrug-resistant bacteria are global health concerns. Reducing the overuse and misuse of antibiotics is the primary step toward minimizing the antibiotic resistance crisis. Thus, it is imperative to introduce and implement novel antimicrobial strategies. Recently, several alternative antimicrobials targeting oxidative stress in bacteria have been studied and shown to be promising. Oxidative stress occurs when bacterial cells fail to detoxify the excessive reactive oxygen species (ROS) accumulated in the cells. Bacteria deploy numerous defense mechanisms against oxidative stress. The oxidative stress response is not essential for the normal growth of bacteria, but it is crucial for their survival. This toxic oxidative stress is created by the host immune response or antimicrobials generating ROS. ROS possess strong oxidation potential and cause serious damage to nucleic acids, lipids, and proteins. Since ROS-based antimicrobials target multiple sites in bacteria, these antimicrobials have attracted the attention of several researchers. In this review, we present recent ROS-based alternative antimicrobials and strategies targeting oxidative stress which might help in mitigating the problem of antibiotic resistance and dissemination.

## 1. Introduction

Antibiotics are used in large quantities in healthcare to treat and even to prevent bacterial infections. However, excessive use of antibiotics has led to the increased development of antibiotic resistance in bacteria [1,2,3]. An increase in antibiotic resistance and its dissemination in hospital and community settings have raised the load on human health. Antibiotic resistance is also a huge economic burden that costs about 20 billion US dollars per year [4]. Antibiotic resistant infections affect more than 2.8 million people and lead to more than 35,000 deaths, every year, in the U.S. alone [5]. The numbers have been predicted to exacerbate to about 10 million deaths worldwide annually by the year 2050 [6] if no drastic measures to reduce the development and spread of antibiotic resistance are taken. Bacterial resistance to antibiotics and the spread especially of the ESKAPE pathogens is concerning. The acronym ESKAPE stands for *Enterococcus faecium*, *Staphylococcus aureus*, *Klebsiella pneumoniae*, *Acinetobacter baumannii*, *Pseudomonas aeruginosa*, and *Enterobacter* spp. [7]. These ESKAPE pathogens can also easily acquire antibiotic resistance through horizontal gene transfer [7]. Hence, there is a pressing need to develop promising alternative antimicrobials and strategies to tackle the antibiotic resistance problem. Minimizing the usage of antibiotics in health care, agriculture, and the environment is key. In addition, it is crucial to develop alternative antimicrobial therapies such as those mediated by reactive oxygen species (ROS). These approaches can help mitigate the problem of antibiotic resistance worldwide [8]. Alternative antimicrobials using ROS as their mechanism of action act simultaneously on several targets such as proteins, lipids, and nucleic acids. The accumulation of ROS causes oxidative stress in bacteria [9,10,11].

## 2. Oxidative Stress in Bacteria

Oxidative stress occurs when prooxidants overpower antioxidants, i.e., ROS get accumulated in a bacterial cell and exceed the cell’s capacity to readily detoxify the ROS [11]. This can happen during the host immune response or due to treatment with an antimicrobial. Oxidative stress can be endogenous or exogenous. Host–pathogen interaction causes exogenous oxidative stress in bacteria, while, intracellular redox reactions, antibiotics, and aerobic respiration contribute to endogenous oxidative stress [9]. Oxygen is the terminal electron acceptor in aerobic respiration and it results in the formation of water upon undergoing complete reduction. However, when oxygen comes in contact with flavoproteins such as oxidases and monooxygenases and undergoes incomplete reduction, it leads to the formation of ROS instead of water [9]. Endogenous ROS such as superoxide anion (O_2_^−^) and hydrogen peroxide (H_2_O_2_) are formed as by-products of aerobic respiration when oxygen interacts with the reduced FAD cofactor of flavoenzymes [9,12]. ROS cause multiple damages to the bacterial cells. They cause double stranded breaks in DNA by oxidizing dCTP and dGTP pools which results in the misincorporation of bases into DNA. Additionally, ROS peroxidate lipids, and carbonylate proteins [13]. Several antimicrobials including some antibiotics generate ROS; these antimicrobials are listed below in Table 1.

## 3. Antibiotics and Oxidative Stress

Antibiotics most commonly target the cell wall (e.g., ampicillin), protein synthesis (e.g., kanamycin), and DNA replication (e.g., norfloxacin) in bacteria as their primary mechanism of action [14,15]. However, some studies reported an intriguing observation that antibiotics generate ROS through overstimulation of electrons via the tricarboxylic acid cycle and the release of iron from the iron–sulfur clusters activating the Fenton chemistry. Thus, antibiotics with unrelated primary modes of action were found to be using a common secondary target by generating ROS [15,16]. Nitrofurantoin and Polymyxin B are two commonly used ROS-mediated antibiotics.

## 4. ROS-Mediated Antibiotics

### 4.1. Nitrofurantoin

Nitrofurantoin is used in the treatment of urinary tract infections typically caused by *E. coli* [11]. The mechanism of action is by its NADH-dependent reduction which generates nitroaromatic anion radicals. Autooxidation of these anion radicals produces O_2_^−^ in the presence of O_2_, thereby exerting the drug’s toxicity on bacteria [11].

### 4.2. Polymyxin B

Antimicrobial peptides such as Polymyxin B (PMB) serve as antibacterial agents. PMB is mainly used to treat infections caused by Gram-negative bacteria such as *A. baumannii, P. aeruginosa*, and carbapenemase-producing Enterobacteriaceae [11,17]. However, it is advised to use PMB only as a last resort drug due to its neurotoxic and nephrotoxic properties [11]. Sampson et al. (2012) showed that PMB induced cell death in Gram-negative bacteria by the accumulation of OH^•^ [11,18].

Oxidative stress induction was first shown to be a part of the mechanism of action of an antibiotic in *Salmonella typhimurium* by Arriaga-Alba and co-workers [19]. Antibiotics were shown to upregulate many oxidative stress genes in *P. aeruginosa*. The induction of antioxidant enzymes on exposure to antibiotics indicated that oxidative stress in bacteria contributed to the lethality of the antibiotic [14,20]. A potential pathway linking ROS (hydroxyl radicals) to antibiotic lethality in *Escherichia coli* was assessed by Wang and Zhao, (2010). They found out that norfloxacin was more lethal in *E. coli* deficient in catalase gene *katG* than in its isogenic mutants [12]. Similarly, ampicillin and kanamycin also showed increased lethality to an alkyl hydroperoxide reductase *ahpC E. coli* mutant. Overall, these investigations indicated that treatment of *E. coli* with antibiotics increased superoxide levels in the bacterium. The superoxide further dismutated to H_2_O_2_ and generated the highly toxic hydroxyl radical which amplified the lethality of antibiotics. According to this study, oxidative stress contributes to the lethal effect of all three classes of tested antibiotics—namely, fluoroquinolones, β-lactams, and aminoglycosides [12].

Some studies like that by Hong and associates (2019) suggest that bacteria exposed to lethal stressors may not die during the actual treatment but post-treatment due to post-stress ROS-mediated toxicity which they confirmed with enzymatic suppression of post-stress ROS. They studied the effect of stressors in *E. coli* with an objective to investigate the role of secondary damage after the removal of the initial stressor [13,21]. It was observed that the ROS levels continued to surge even after the removal of the stressor leading to cell death caused by post-stress ROS. When they blocked ROS accumulation using external agents, the post-stress cell death was either inhibited or slowed down [13].

The reports on the connection between antibiotics and ROS are contradictory. Some reports [22,23,24] disagree that the generation of ROS contributes to the lethality of antibiotics as antibiotics also seem to work under anoxic conditions. Despite the contradiction in both theories, it is apparent that bacterial metabolism influences antibiotic lethality, for example, iron homeostasis and iron-sulfur proteins affect antibiotic lethality [14,15,22,25]. Although, it is still unclear whether ROS influence antibiotic lethality, numerous alternative antimicrobials induce oxidative stress in bacteria. The mechanism of action of these antimicrobials is predominantly mediated by the generation of ROS.

## 5. ROS-Based Alternative Antimicrobials

Antibiotics’ efficacy is in danger because of the increasing bacterial drug resistance [13]. However, there are several novel alternative antimicrobials under development whose primary mode of action is through the generation of ROS causing oxidative stress in bacteria. These antimicrobials often target the redox defenses such as the thiol-dependent enzyme thioredoxin reductase (TrxR) in bacteria [4,26].

Advances in the development of ROS-mediated antimicrobials have widened the avenues of alternative antimicrobials beyond metals and nanoparticles. Some examples include ebselen, nanoparticles, nanozymes, AGXX^®^, natural compounds like allicin and surgical honey, antimicrobial photodynamic therapy (aPDT), and non-thermal plasma (NTP) [11,27,28].

### 5.1. Ebselen

Ebselen is an organoselenium-based antioxidant drug that has anti-inflammatory, antioxidant, and cryoprotective properties. It inhibits TrxR in bacteria and this inhibition results in oxidative stress. Ebselen was recently shown to efficiently inhibit the growth of multi-drug resistant *S. aureus* [26]. Dong and co-workers also tested the efficacy of Ebselen as a topical drug in rat models. They reported significant improvement in wound healing in Ebselen-treated rats as compared with the control rats. The treatment with the drug also reduced the bacterial load in *S. aureus* skin lesions in rats [26]. Ebselen has also been used in combination with ROS-producing antimicrobials such as silver nanoparticles [4].

### 5.2. Nanoparticles

Nanoparticles are <100 nm in size. They generate high amounts of ROS which cause damage to living cells by carbonylation of proteins, peroxidation of lipids, DNA/RNA breakage, and membrane structure destruction [29]. Nanoparticles made of silver, silver oxide, titanium dioxide, silicon, copper oxide, zinc oxide, gold, calcium oxide, and magnesium oxide have been studied and found to be effective against both Gram-positive and Gram-negative bacteria [30].

Metals like silver and copper have been used as antimicrobials since ancient times [31]. Diaz-Garcia and co-workers conducted a study where they synthesized mesoporous silica nanoparticles (MSNs) containing a maleamato ligand (MSN-maleamic) followed by coordination of copper (III) ions (MSN-maleamic-Cu) as an antibacterial agent [32]. They tested the effect of their antibacterial on *E. coli* and *S. aureus* (MSN-maleamic-Cu was tested) revealing that MSN-maleamic and MSN-maleamic-Cu both triggered oxidative stress mechanisms in both Gram-positive *S. aureus* and Gram-negative *E. coli* [32]. The minimum inhibitory concentration (MIC) of MSN-maleamic and MSN-maleamic-Cu were determined by performing microdilution assays. Both antimicrobial preparations showed higher antibacterial activity against *E. coli* as compared with *S. aureus*. Their data indicate that MSN-maleamic (MIC = 62.5 µg/mL) works more efficiently than MSN-maleamic-Cu (MIC = 125 µg/mL) in case of *E. coli.* In *S. aureus*, the MIC for MSN-maleamic was 250 µg/mL, and >250 µg/mL in the case of MSN-maleamic-Cu [32]. *S. aureus* treated with MSN-maleamic generated 50% more ROS than the untreated *S. aureus* while *S. aureus* treated with MSN-maleamic-Cu generated 30% more ROS as compared with the untreated *S. aureus*. In *E. coli*, exposure to both MSN-maleamic and MSN-maleamic-Cu led to the generation of 40% more ROS as compared with the untreated *E. coli* [32]. ROS generation values corresponded to the MIC values of the antibacterials. MSN-maleamic produced more ROS and had a lower MIC as compared with MSN-maleamic-Cu. This indicates that ROS generation or oxidative stress might contribute to the antibacterial mechanism of action of MSN-maleamic and MSN-maleamic-Cu [32]. Besides nanoparticles, other nanomaterials such as those possessing enzyme-like characteristics are also in use. These nanomaterials are termed nanozymes [33].

### 5.3. Nanozymes

Nanozymes produce ROS and are thus promising antimicrobials [33]. One of the disadvantages of ROS is that they cannot distinguish between bacterial and mammalian cells. However, recent studies have shown that nanozymes that produced surface-bound ROS killed bacterial cells over mammalian cells [33]. The surface-bound ROS on silver-palladium bimetallic alloy (AgPd0.38) not only efficiently killed antibiotic-resistant bacteria but also delayed the development of drug resistance [33]. In addition, AgPd0.38 also inhibited biofilm formation [33]. Toxic ROS have <200 nm effective radii of action and have short lifetimes. According to this study, if nanozymes produce surface-bound ROS, these ROS kill bacterial cells over mammalian cells. This is based on the fact that nanoparticles remain extracellular if they are above a certain size. Bacteria cannot take nanoparticles up that are larger than 4 nm for gold nanoparticles and 40 nm for silver nanoparticles unless the bacterial cell wall is impaired. However, mammalian cells take nanoparticles up via endocytosis which traps the nanoparticles in endocytic vesicles. Damage to these vesicles does not affect the cell lethally [33].

Several metal-based antimicrobials in addition to the silver-palladium bimetallic alloy discussed above have been studied. One such metal-based novel antimicrobial is AGXX^®^.

### 5.4. AGXX^®^

AGXX^®^ is an antimicrobial surface coating consisting of micro galvanic elements of silver and ruthenium. It can be coated or deposited on various carriers such as cellulose, plastics, ceramics, or metals. AGXX^®^ is only slightly cell-toxic [34] and no resistance towards AGXX^®^ has been reported so far [35]. AGXX^®^ catalytically produces ROS and has been shown to efficiently inhibit the growth of pathogens like *Enterococcus faecalis*, and methicillin-resistant *S. aureus* (MRSA). AGXX^®^ also inhibited biofilm formation in MRSA [36,37]. Exposure to AGXX^®^ induced general stress, heat shock (expression of Clp proteases), copper stress, and oxidative stress response in both *E. faecalis* and MRSA [36,37]. AGXX^®^ also interfered with iron homeostasis and upregulated several siderophore biosynthesis (*sbn*) genes in MRSA [37,38]. In *S. aureus*, AGXX^®^ also caused increased protein-thiol-oxidations, protein aggregations, and an oxidized bacillithiol (BSH) redox state [35,38]. The mechanism of action has been summarized in Figure 1.

As presented in Figure 1, AGXX^®^ affects oxidative stress defenses such as superoxide dismutase (SodA), catalase (KatA), alkyl hydroperoxide reductase (AhpCF), thioredoxin/thioredoxin reductase (Trx/TrxR), MerA, and BSH. The multiple modes of action of AGXX^®^ reduce the likelihood of the development of resistance in bacteria and give AGXX^®^ an advantage over conventional antimicrobials [38].

Apart from synthetic compounds, ROS-mediated natural compounds like allicin and honey have also been reported for their efficient antimicrobial activity.

## 6. Natural Compounds

Allicin (from garlic) is a thiol-reactive compound; it decreases the levels of low molecular weight thiol which functions as a defense mechanism against ROS, thereby reducing the oxidative stress caused by ROS [39].

Surgihoney Reactive Oxygen (SHRO) is a pharmaceutical honey wound gel [8]. It uses glucose oxidase in medicinal grade honey to oxidize sugar which leads to the generation of H_2_O_2_ [40]. When applied to a wound, SHRO provides a constant level of ROS over a long period. SHRO has been used in the treatment of soft tissue infections caused by MRSA and *P. aeruginosa* and also exhibits antibiofilm activity [8].

In addition to natural compounds, alternative approaches including using photodynamic therapy and non-thermal plasmas have caught attention of researchers.

## 7. Photodynamic Therapy

Recently, aPDT has been used to eliminate bacterial biofilms. aPDT uses photosensitizers (PS) which are non-toxic dyes that can be excited by harmless visible light to generate ROS [27]. The aPDT approach involves three steps: (i) topical administration of PS, (ii) light irradiation, and (iii) interaction of the excited state of PS with oxygen [27]. ROS are generated upon photoactivation and damage proteins, lipids, and nucleic acids in the biofilm matrix, inside and on the surface of the cells [27]. However, since ROS are non-specific, they end up damaging both planktonic and biofilm cells [27]. Two oxidative mechanisms of photoinactivation are involved in the inactivation of bacterial cells. In the type I pathway, the electrons/hydrogen from the PS triplet excited state interact with a substrate to produce radical ions. In the triplet excited state, the excited electron is not paired with the ground state but is parallel to the ground state. In the type II pathway, the energy from that triplet state is transferred to molecular oxygen to produce singlet oxygen. Both type I and II pathways generate highly toxic ROS [41]. The authors studied the effect of aPDT on *Vibrio fischeri* and recombinant *E. coli*. They used 5,10,15-tris(1-Methylpyridinium-4-yl)-20-(pentafluorophenyl)-porphyrin triiodide (Tri-Py+-Me-PF) as PS to determine the recovery of the viability of bacteria and the development of resistance in bacteria. To investigate the bacterial recovery, bacteria were exposed to 5.0 µM Tri-Py+-Me-PF and were irradiated with white light for 270 min. The bacteria were then protected from the light, aliquots were collected, and bioluminescence was measured. To check if the bacteria develop any resistance toward the treatment, bacteria were exposed to 5.0 µM Tri-Py+-Me-PF and white light for 25 min. The colonies which survived the first irradiation were re-suspended in PBS. The protocol was repeated 10 times. The results suggested that aPDT efficiently damaged *V. fischeri* and *E. coli*. Neither of the bacteria recovered their viability after treatment, nor did they develop any resistance towards the treatment [41].

## 8. Non-Thermal Plasma

NTPs are plasmas with particles that are not in thermal equilibrium [28]. Cold atmospheric plasma is another type of plasma that is used as an antimicrobial [28]. It is produced by bioelectric discharges or plasma jets [42]. Kvam et al. studied the effect of NTP on drug-resistant clinical isolates of MRSA and *P. aeruginosa* [43]. Direct fluorescent imaging showed that NTP efficiently and quickly (>5 orders of magnitude in 30 s) inactivated planktonic cultures. The inactivation occurs through damage to the cell surface which leads to the loss of membrane integrity, leakage of nucleic acids, proteins, and ATP, and finally with longer exposure time to the focal dissolution of the cell surface [43].

Oxidative stress imposed by antimicrobials can cause serious damage to bacterial cells. To overcome oxidative stress, bacteria use their defense systems [11].

## 9. Bacterial Response to Oxidative Stress

Bacteria circumvent oxidative stress using an arsenal of defenses such as detoxifying enzymes like catalase, alkyl hydroperoxide reductase, thioredoxin, and superoxide dismutase; pigments like carotenoids; metal homeostasis, DNA repair, general stress response, and SOS response [9,11]. These detoxification and repair mechanisms are regulated by gene networks [9]. Bacterial response to ROS and the regulation of gene networks that overcome oxidative stress in bacteria have been comprehensively reviewed in previous articles [9,44]. Bacterial response to ROS is regulated by transcription factors such as OxyR, PerR, OhrR, and SoxRS. These regulons control the expression of genes associated with antioxidant defense, genes like superoxide dismutase, catalase, thioredoxins, heme biosynthesis machinery, glutathione reductases, ferric uptake regulator, and bacterioferritin. OxyR is a transcriptional regulator which upregulates genes from its regulon to detoxify ROS in response to H_2_O_2_ accumulation [11]. When bacteria encounter excessive superoxide, the cells increase the transcription of genes such as *sodA* and *sodB* which produce superoxide dismutase to protect the cells from the lethal effect of ROS. These genes are regulated by SoxRS. Exposure to H_2_O_2_ activates the OxyR regulon, which regulates the expression of protective genes such as *katG* and *ahpC* 15 [40]. Iron homeostasis and remodeling of metabolism are two other factors that play a pivotal role in alleviating the damage caused by ROS. Some examples of remodeling of metabolism are: (i) upregulation of the glycoxylate shunt which reduces endogenous ROS formation, (ii) rerouting the metabolism toward the pentose phosphate pathway and enhancing production of the cofactor NADH which replenishes the level of antioxidants. Similarly, ketoacids (pyruvate and α-ketoglutarate) can undergo decarboxylation in the presence of ROS. This step generated toxic by-products and hence diminishes the damage caused by ROS [44].

Iron is crucial for the growth and survival of bacteria [45]. It is also involved in the generation of ROS via Fenton reaction and in oxidative stress. Under iron starvation conditions, bacteria produce siderophores to facilitate iron uptake [46]. Siderophore production or siderophore-mediated iron acquisition is crucial to combat oxidative stress in bacteria [45]. In *S. aureus*, two siderophores Staphyloferrin A and B were found to enhance resistance to oxidative stress [45]. Similarly, in *E. coli*, the siderophore enterobactin alleviates oxidative stress [45,46]. Oxidative stress can also regulate siderophore production in bacteria [46]. Peralta and co-workers tested the effect of H_2_O_2_ and paraquat on *E. coli* to investigate siderophore-mediated regulation of oxidative stress. Enterobactin reduced the sensitivity of *E. coli* to both H_2_O_2_ and paraquat and the expression of enterobactin increased in *E. coli* when cells were exposed to H_2_O_2_ and paraquat even when iron was present in excess. The siderophore potentially neutralizes ROS [46]. Similar trends were observed in MRSA on exposure to the antimicrobial surface coating AGXX^®^. Siderophore biosynthesis genes were highly upregulated when the bacterium was exposed to AGXX^®^ [37]. As AGXX^®^ primarily acts by the generation of ROS, the induction of *sbn* genes (encoding siderophores) could likely be a response generated by MRSA to neutralize ROS.

Oxidative stress is a significant stress encountered by bacteria. An overview of oxidative stress in bacteria, the damage caused by the ROS, and the repair mechanisms used by bacteria to combat oxidative stress are shown below (Figure 2).

Oxidative stress response in bacteria may protect them from the host immune response as well as from the antibiotic treatment. This facilitates bacteria to develop persistent infections [47]. Persisters are bacterial cells in a dormant state with low metabolic activity. These cells showcase high antibiotic tolerance and can recolonize post-therapy [48]. Persisters are less sensitive to ROS as compared with their antibiotic susceptible counterparts; an increased expression of efflux pumps is the proposed defense mechanism. Efflux pumps are a mechanism of oxidative stress response and they likely act by enabling the efflux of ROS-damaged proteins [47]. ROS can increase persister levels through a decrease in membrane potential and reduced metabolism. Surveillance of ROS levels is crucial to defend persister formation [48].

## 10. Conclusions

In summary, although using ROS-generating antimicrobials and novel strategies can be considered promising, there are some limitations to this approach. ROS cannot distinguish between bacterial and mammalian cells, and since they directly damage DNA, they can induce mutations in specific targets of antibiotics, thereby leading to antibiotic resistance and establishment of persistence [44,45]. Further research on ROS-based antimicrobials and antimicrobials acting simultaneously on multiple targets is crucial for managing antibiotic resistance in bacteria.

## Figures and Tables

**Figure 1 microorganisms-10-00061-f001:**
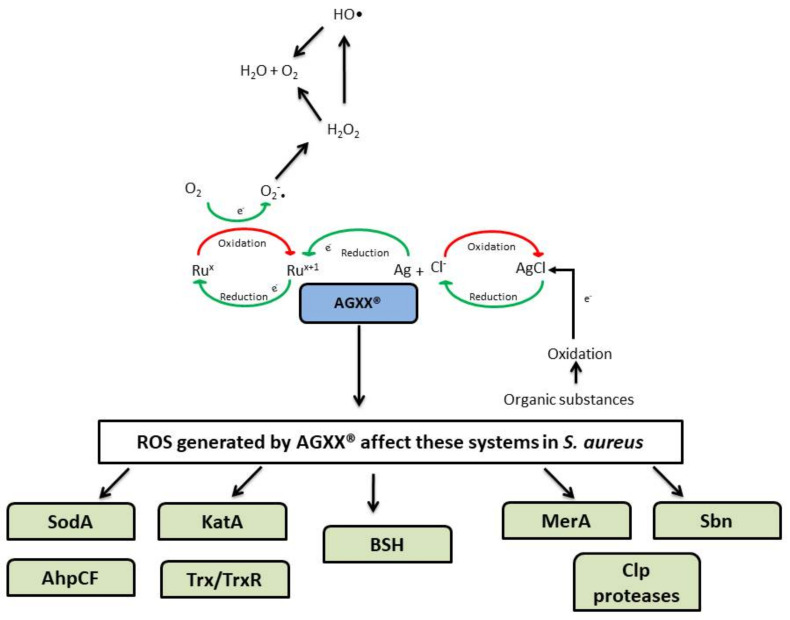
Mechanism of action of AGXX^®^ in *S. aureus*. The mode of action of AGXX^®^ is based on two inter-connected redox cycles resulting in its antimicrobial effect and self-renewal. In the first cycle, the chloride (Cl^−^) from the electrolyte oxidizes elementary silver (Ag^0^) to form silver chloride (AgCl). Subsequently, oxidation of organic matter like sugars in bacteria leads to the reduction of AgCl to form Ag and Cl^−^. In the second cycle, Ru^x+1^ is reduced to Ru^x^ and reoxidized to its initial state (x is used because the precise oxidation state of ruthenium is unknown). Reoxidation of Ru^x^ to Ru^x+1^ results in the formation of ROS such as superoxide anion, hydrogen peroxide, and highly toxic HO•. These redox cycles facilitate the constant regeneration of AGXX^®^. This figure has been Adapted with permission from ref. [36]. 11 November 2021, Elsevier and from ref. [35].

**Figure 2 microorganisms-10-00061-f002:**
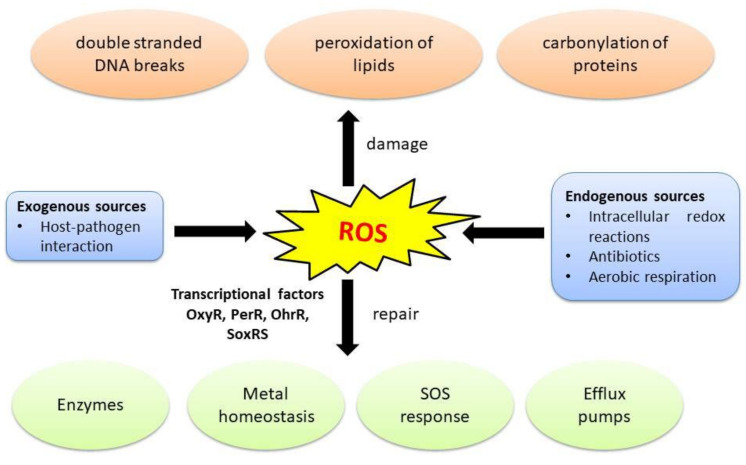
Overview of oxidative stress in bacteria and their response to the stress. Accumulation of ROS causes oxidative stress in bacteria. Both exogenous and endogenous sources contribute to oxidative stress. ROS damage bacterial cells by causing double stranded DNA breaks, peroxidation of lipids, and by carbonylation of proteins. Bacteria respond to oxidative stress using a battery of repair mechanisms such as production of protective enzymes, metal homeostasis, SOS response, and efflux pumps. Oxidative stress response in bacteria is regulated by transcriptional factors OxyR, PerR, OhrR, and SoxRS.

**Table 1 microorganisms-10-00061-t001:** List of ROS-mediated alternative antimicrobials/methods generating alternative antimicrobials.

• **ROS-mediated antibiotics**
Nitrofurantoin
Polymyxin B
• **ROS-mediated antimicrobials**
Ebselen
Nanoparticles
Nanozymes
AGXX^®^
• **Natural compounds**
Allicin
Surgihoney
• **Photodynamic therapy**
• **Non-thermal plasma**

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
