# Peer review of "Antimicrobials Functioning through ROS-Mediated Mechanisms: Current Insights"

_microorganisms, 2021, doi:10.3390/microorganisms10010061_

Round 1

Reviewer 1 Report

Antimicrobials functioning to ROS   Manuscript ID: microorganisms-1508541

Antimicrobial resistance is a great health concern globally. This review is timely. The information provided in this review includes new strategies to combat antimicrobial resistance through ROS generation. This review summarizes recent ROS based antimicrobials.The review is well organized, well written and easy to follow. Most of the references include citations within the past 5 years. I have one suggestion for the authors. It will be nice to present a Table, which summarizes the 1) ROS mediated antimicrobials (Ebselen, metal nanoparticles, nanonezymes and the AGXX) 2) the ROS mediated antibiotics (nitrofurantoin and polymyxin B) 3) natural compounds (Allicin and surgihoney) and 4) the photodynamic therapy and the NTP plasma.

Author Response

We thank the reviewer for the positive comments and suggestions. We have added a list summarizing all the ROS-mediated antimicrobials discussed in the review as suggested by the reviewer. The list has been added in line 66.

Reviewer 2 Report

A very interesting topic of manuscript and definitely because of  increasing resistance to antibiotics, new alternative methods are necessary and oxidative stress is definitely important. My main complaint about the manuscript is the lack of clarity and I didn't get along while reading the paper. It would be good to separate all antibiotics under one title and present them with a diagram or table with a mechanism of action.
This is followed by ROS based alternative antimicrobials and it is not clear to me why nitrofurantoin is described here. Furthermore, this part of should be divided into subheadings for easier reading. The conclusion should definitely be dissociated as it turns out that part of the defensive response of bacteria to oxidative stress.
 Also, it is common to write the full name of bacteria at the first mention of bacteria and only later an abbreviated form. Check the correct name of Salmonella Typhimurium.

Author Response

We thank the reviewer for the constructive comments and suggestions. We have separated all the antibiotics under one title and have presented the antibiotics in a list for better clarity. We have divided the sections discussing ROS-mediated antimicrobials into subheadings for easier reading as suggested by the reviewer. We have added a separate section for the conclusion as suggested by the reviewer. We have checked the correct name of Salmonella Typhimurium in the reference.

Reviewer 3 Report

In the manuscript ID: microorganisms-1508541 the authors do an overview about the oxidative stress as a novel mechanism adoptable for the development of new antimicrobial strategies. Starting from the description of the oxidative stress in the bacterial cell, they report the antibiotic contribution to its induction and the new ROS-based antimicrobials (compounds, metals and peptides); finally, they describe the bacterial response to the oxidative stress itself.

Overall, the paper is clear, interesting and provides new insights on novel antimicrobial strategies, which are warranted considering the rise of antibiotic resistance.

However, there are some concerns that should be addressed before publication:

-As the development of subpopulation of bacterial persisters has been indicated to be potentiated under oxidative stress (doi: 10.4161/viru.23987; doi: 10.1038/srep43839), this topic should be added in the last section “Bacterial response to oxidative stress”.

-Moreover, further figures should be added to the manuscript, representing the main parts of the review: for example, one figure about oxidative stress in the bacterial cell and the antibiotic contribution; another figure about bacterial response to oxidative stress.

After these major revisions, the paper can be considered for publication in “Microorganisms”.

MINOR COMMENTS

Line 70, do the authors mean “an antimicrobial effect”?

Line 74, please use the full name “Escherichia coli”;

Lines 130-132, please rephrase “S. aureus treated with MSN-maleamic and MSN-maleamic-Cu generated 50% and 30%, respectively, more ROS than the untreated control” to make the sentence clearer;

Line 145, “in bacteria” can be deleted.

Author Response

We thank the reviewer for his insightful comments and suggestions. We have added a short section on bacterial persisters in the last section “Bacterial response to oxidative stress” as suggested by the reviewer. We have added a new figure to the manuscript depicting oxidative stress in bacteria and their response to oxidative stress. The figure provides an overview of oxidative stress in bacteria, damage caused by the reactive oxygen species, and the strategies used by bacteria to overcome the oxidative stress. 

MINOR COMMENTS

Line 70, do the authors mean “an antimicrobial effect”?

Authors’ response: We have rephrased this sentence for better clarity (line 90).

Line 74, please use the full name “Escherichia coli”;

Authors’ response: We have used the full name Escherichia coli in the revised manuscript (line 95).

Lines 130-132, please rephrase “S. aureus treated with MSN-maleamic and MSN-maleamic-Cu generated 50% and 30%, respectively, more ROS than the untreated control” to make the sentence clearer;

Authors’ response: We have rephrased the sentence in the revised manuscript (lines 170-172).

Line 145, “in bacteria” can be deleted.

Authors’ response: We have deleted “in bacteria” in the revised manuscript (line 188).

Round 2

Reviewer 2 Report

I am satisfied with the responses and corrections provided by the authors.

Reviewer 3 Report

In the revised version of the manuscript ID: microorganisms-1508541 the authors have successfully addressed the raised concerns, implementing the paper with the required information and making it clearer for the readers. The paper can now be published in “Microorganisms”.